

# Meta-analysis of commonly mutated genes in leptomeningeal carcinomatosis

Irem Congur[1,*], Ekin Koni[1,*], Onur Emre Onat[2,3] and
Zeynep Tokcaer Keskin[1,4]

[1] Department of Molecular and Translational Biomedicine, Institute of Natural and Applied Sciences, Acıbadem Mehmet Ali Aydınlar University, Istanbul, Turkey
[2] Department of Genome Studies, Institute of Health Sciences, Acıbadem Mehmet Ali Aydınlar University, Istanbul, Turkey
[3] Department of Molecular Biology, Institute of Life Sciences and Biotechnology, Bezmialem Foundation University, Istanbul, Turkey
[4] Department of Molecular Biology and Genetics Faculty of Engineering and Natural Sciences, Acıbadem Mehmet Ali Aydınlar University, Istanbul, Turkey
* These authors contributed equally to this work.

Corresponding author
Zeynep Tokcaer Keskin,
zeynep.keskin@acibadem.edu.tr

## ABSTRACT

**Background:** Leptomeningeal carcinomatosis (LMC) is a rare type of cancer that settles at the meninges through metastasis of non-small cell lung cancer (NSCLC), breast cancer and melanoma. The molecular mechanism underlying LMC is not known, therefore molecular studies investigating the development of LMC are needed. Here, we aimed to identify commonly mutated genes in LMC caused by NSCLC, breast cancer, and melanoma using an in-slico approach and their interactions using integrated bioinformatic approaches/tools in this meta-analysis.

**Methods:** We conducted a meta-analysis using information from 16 studies that included different sequencing techniques of patients with LMC caused by three different primary cancers: breast cancer, NSCLC, and melanoma. All studies that assessed mutation information from patients with LMC were searched in PubMed, from their inception to February, 16 2022. Studies that performed NGS on LMC patients with NSCLC, breast cancer, or melanoma were included, while studies that did not apply NGS to CSF samples, did not provide information on altered genes, were reviews, editorials, or conference abstracts, or whose main goal was the detection of malignancies were all excluded. We identified commonly mutated genes in all three types of cancer. Next, we constructed a protein-protein interaction network, then performed pathway enrichment analysis. We searched National Institutes of Health (NIH) and Drug-Gene Interaction Database (DGIdb) to find candidate drugs.

**Results:** We found that *TP53, PTEN, PIK3CA, IL7R*, and *KMT2D* genes were commonly mutated genes in all three types of cancer *via* our meta-analysis that consisted out of 16 studies. Our pathway enrichment analysis showed that all five genes were primarily associated with regulation of cell communication and signaling, and cell proliferation. Other enriched pathways included regulation of apoptotic processes of leukocytes and fibroblasts, macroautophagy and growth. According to our drug search we found candidate drugs; Everolimus, Bevacizumab and Temozolomide, which interact with these five genes.

**Conclusion:** In conclusion, a total of 96 mutated genes in LMC were investigated *via* meta-analysis. Our findings suggested vital roles of *TP53, PTEN, PIK3CA, KMT2D*,

and *IL7R*, which can provide insight into the molecular basis of LMC development and paving the door to the development of new targeted medicine and will encourage molecular biologists to seek biological evidence.

## INTRODUCTION

Leptomeningeal carcinomatosis (LMC) is a rare type of cancer that settles through metastasis from a tumor in the body to the meninges. The leptomeninges are the brain and spinal cord areas that contain cerebrospinal fluid (CSF). Metastatic cells involved in CSF take over the central nervous system in a short time and affect the brain, spinal cord, and nerves, causing sudden neurological disorders and death. Cancerous cells reach the brain through a vein and migrate from the choroid plexus to the CSF. Entering the leptomeningeal region, the metastatic cancer cell can circulate freely, thereby causing extensive damage to the nervous system and brain (*Boire et al., 2017*).

The mortality rate in LMC, which follows an aggressive path, is also quite high. Most solid tumors are known to cause LMC, but the most common solid tumors involving leptomeninges include breast, lung, and melanoma. The incidence of cancer varies between different types of cancer. It is 5% to 8% among breast cancer patients (*Wang, Bertalan & Brastianos, 2018*) and is the most common type of fatal cancer (*Li et al., 2018a*); it ranges from 9% to 25% in lung cancer and 6% to 18% in melanomas. Radiation therapy and systemic intrathecal or intravenous chemotherapy methods are used for the treatment of LMC (*Chahal et al., 2015*). But the average life expectancy of tumors on the membranes of the meninges with the prescribed treatments is an average of 6 months, despite the longer survival time of some patients (*Wang, Bertalan & Brastianos, 2018*; *Özdoğan & Çoban, 2004*). Within the framework of the unknown molecular mechanism of the disease, next-generation sequencing (NGS) is being performed to find out the mutation status of circulating tumor cells (CTCs) found in CSF and RNA sequencing (RNA-seq) is being performed to investigate the transcriptome properties (*Ruan et al., 2020*). Currently, the diversification of cancer treatment and the prolongation of patient survival have also begun to increase LMC incidence (*Lee et al., 2013*). Therefore, molecular studies investigating the development of LMC are needed. The aim of our meta-analysis is to explore commonly mutated genes and their interactions in LMC caused by NSCLC, breast cancer, and melanoma with integrated bioinformatic approaches.

In this meta-analysis, 16 scientific articles that performed different sequencing techniques to LMC patients caused by breast cancer, NSCLC, or melanoma were included. Mutated gene lists were created with the given information in the detailed analysis on their genetic information obtained from results and supplementary information, which were available online. We categorized and compared mutated genes of three types of primary cancers that caused LMC and found that *TP53, PIK3CA, PTEN, KMT2D* and *IL7R* were
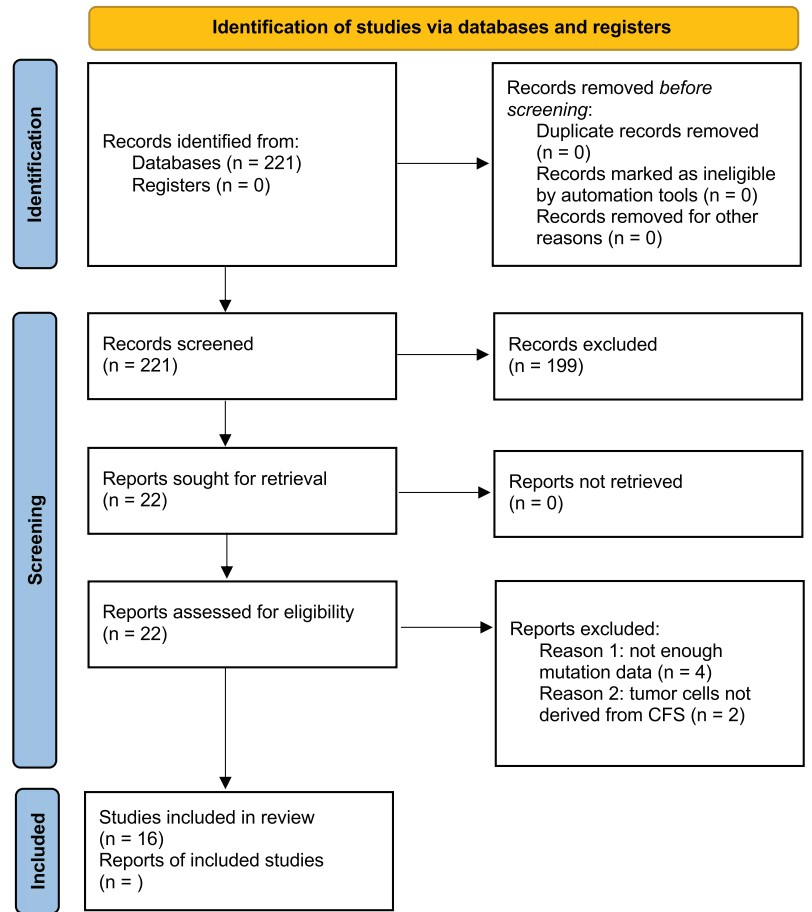

**Figure 1** PRISMA 2020 flow diagram of the study selection.

commonly mutated in all three types of cancers. To reveal protein interactions, we constructed a protein-protein interaction network and performed pathway enrichment analysis. Our results suggested several key pathways such as regulation of cell communication and signaling, and cell proliferation might have a role in LMC development and three candidate drug options, Everolimus, Bevacizumab and Temozolomide, that can be used for LMC treatment.

# MATERIALS AND METHODS

The Guideline for Preferred Reporting Items for Systematic Reviews and Meta-Analyses (PRISMA) statement was followed in the conduct and reporting of this meta-analysis (*Page et al., 2021*).

## Literature search

We carried out a literature search on PubMed database using Medical Subject Headings (MeSH) indexing terms and keywords associated with leptomeningeal carcinoma (File S1). Without any additional publication restrictions, the search was carried out in English. This analysis covered articles published between February 16, 2012, and February 16, 2022 that

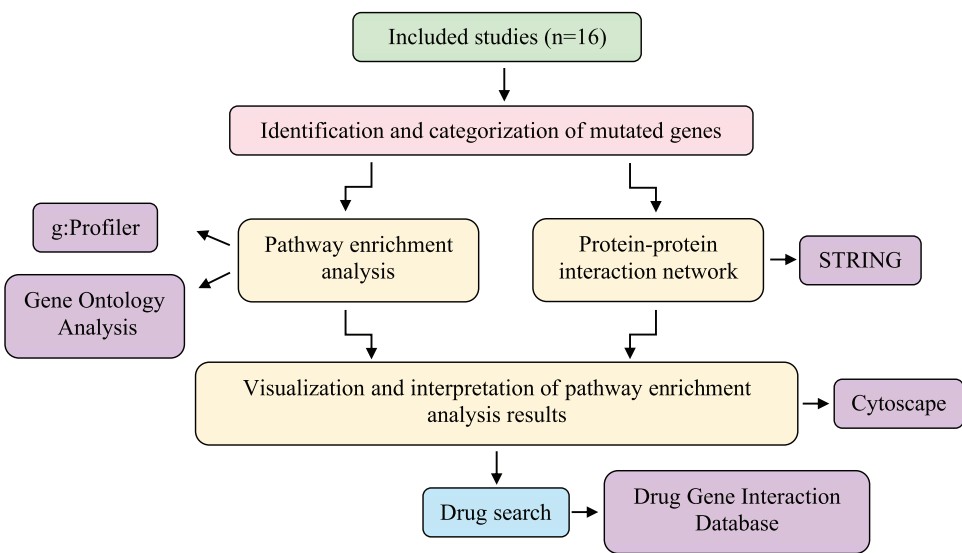

**Figure 2 Systematic workflow of the methodology.** The meta-analysis was conducted using information obtained from 16 studies involving different sequencing techniques of patients with LMC caused by three different primary cancers (breast cancer, NSCLC, and melanoma). Then, a protein-protein interaction network was created, and pathway-enrichment analysis was performed. Candidate drugs were investigated *via* the Drug-Gene Interaction Database and interpreted with the genes obtained as a result of the analysis. LMC, leptomeningeal carcinomatosis; NSCLC, non-small cell lung cancer.

provided information on mutated genes among patients with LMC. This time frame was chosen to include more modern studies, reflecting the significance of sequencing technologies. As part of the review process, references from publications that were included were also scanned; however, a search of the grey literature was not performed.

## Inclusion and extraction criteria

Clinical studies that performed NGS to LMC patients caused by NSCLC, breast cancer, or melanoma were included.

Studies were excluded if they did not apply NGS to CSF samples, lacked to provide accurate information regarding mutated genes, were reviews, editorials, or conference abstracts, or if their primary intention was the detection of tumors. Studies with objectives that did not entail using human patients were also excluded after evaluation (Fig. 1). An analysis path has been followed as described in the following steps and shown in Fig. 2.

## Data selection

Since the data and content used in our study are derived from the tables and figures that have already been interpreted in the articles, no loss, modification or editing has been carried out in our results.

The type of article, the title, and the abstract were checked in order to find studies from the literature search that might be relevant. Two reviewers (IC and EK) independently evaluated the entire texts of the articles for pertinent data and then selected which articles to include. The following information was extracted: first author, primary tumor site,

**Table 1** Articles included in the current study.

| Study | Author | Primary tumor location |
|---|---|---|
| Recurrently mutated genes differ between leptomeningeal and solid lung cancer brain metastases | *Li et al. (2018a)* | NSCLC |
| Detection of driver and resistance mutations in leptomeningeal metastases of NSCLC by next-generation sequencing of cerebrospinal fluid circulating tumor cells | *Jiang et al. (2017)* | NSCLC |
| Unique genetic profiles from cerebrospinal fluid cell-free DNA in leptomeningeal metastases of EGFR-mutant non-small-cell lung cancer: a new medium of liquid biopsy | *Li et al. (2018b)* | NSCLC |
| Unique genomic profiles obtained from cerebrospinal fluid cell-free DNA of non-small cell lung cancer patients with leptomeningeal metastases | *Ying et al. (2019)* | NSCLC |
| Unique genomic alterations of cerebrospinal fluid cell-free DNA are critical for targeted therapy of non-small cell lung cancer with leptomeningeal metastasis | *Wang et al. (2021)* | NSCLC |
| Circulating tumor cell characterization of lung cancer brain metastases in the cerebrospinal fluid through single-cell transcriptome analysis | *Ruan et al. (2020)* | NSCLC |
| Different next-generation sequencing pipelines based detection of tumor DNA in cerebrospinal fluid of lung adenocarcinoma cancer patients with leptomeningeal metastases | *Ge et al. (2019)* | NSCLC |
| Detection of circulating tumor DNA from non-small cell lung cancer brain metastasis in cerebrospinal fluid samples | *Ma et al. (2020)* | NSCLC |
| Clinical utility of cerebrospinal fluid cell-free DNA as liquid biopsy for leptomeningeal metastases in ALK-rearranged NSCLC | *Zheng et al. (2019)* | NSCLC |
| Brain tumor mutations detected in cerebral spinal fluid | *Pan et al. (2015)* | NSCLC |
| Cell-cycle and DNA-damage response pathway is involved in leptomeningeal metastasis of non–small cell lung cancer | *Fan et al. (2018)* | NSCLC |
| Genotyping tumour DNA in cerebrospinal fluid and plasma of a HER2-positive breast cancer patient with brain metastases | *Siravegna et al. (2017)* | Breast cancer |
| Clinical significance of detecting CSF-derived tumor cells in breast cancer patients with leptomeningeal metastasis | *Li et al. (2017)* | Breast cancer |
| Cerebrospinal fluid-derived circulating tumour DNA better represents the genomic alterations of brain tumours than plasma | *De Mattos-Arruda et al. (2015)* | Breast cancer |
| Tumor DNA in cerebral spinal fluid reflects clinical course in a patient with melanoma leptomeningeal brain metastases | *Li et al. (2016)* | Melanoma |
| Evaluating circulating tumor DNA from the cerebrospinal fluid of patients with melanoma and leptomeningeal disease | *Ballester et al. (2018)* | Melanoma |

sequencing technique, mutated genes and their patient relevance. Whenever there was a difference of opinion during the screening and extraction processes, a solution was found through discussion and with the advice of a third reviewer (ZTK).

## Identification and categorization of mutated genes

A total of 16 articles were categorized based on their primary tumor location: 11 NSCLC-LMC, three breast-LMC and two melanoma-LMC (Table 1). The causes of the formation of the three types of primary cancer and the prevalence in groups contain differences within themselves. Therefore, the detailed characteristic information of the patients (age, gender, primary tumor mutation, treatment before LMC diagnosis and NGS) have been neglected in the selected studies, whereas the mutated genes detected by NGS applied to CSF samples of LMC patients were included. This information was obtained from "results"

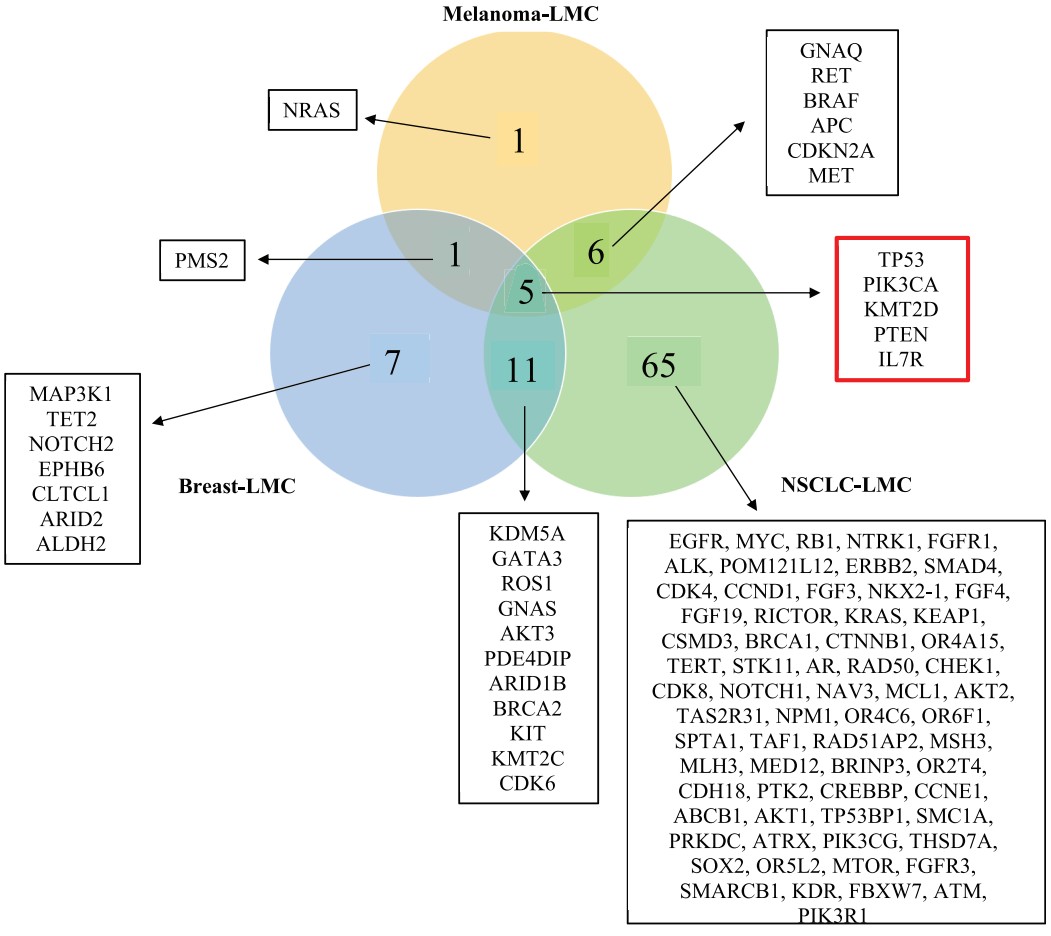

**Figure 3 Venn diagram showing mutated genes in LMC patients according to their primary tumor sites.** When all the genes reported were plotted in a Venn diagram, five genes appeared to be in common. These five genes shown in the red box were *TP53, PTEN, PIK3CA, KMT2D* and *IL7R* were mutated commonly in all three primary tumor sites (NSCLC, breast cancer, melanoma). LMC, leptomeningeal carcinomatosis; NSCLC, non-small cell lung cancer.

or "supplementary information" of each article. Mutated genes observed in only one patient were discarded. Further analysis was performed with the mutated genes ($n \geq 2$, $n$ = patient number) shown in Fig. 3.

## Construction of protein-protein interaction (PPI) network and pathway enrichment analysis

### PPI network construction
For each category of mutated genes (Fig. 3), PPI network was constructed using STRING database (http://string-db.org/) (*Szklarczyk et al., 2019*) version: 11.5. Within basic settings, Full STRING network and confidence boxes were checked.

### Pathway enrichment analysis
g:Profiler (https://biit.cs.ut.ee/gprofiler/gost) (*Raudvere et al., 2019*) version e105_eg52_p16_e84549f (database updated on 03/01/2022) was used for pathway

enrichment analysis. The protocol of *Reimand et al. (2019)* was followed. Within the options section, the following boxes were checked: *Homo sapiens* and ordered query. According to *Reimand et al. (2019)* within the options section, after selecting *Homo sapiens* as a specific organism, the ordered query box was checked. Then, for GO analysis, the biological process (GO: BP) section and for the biological pathways section the Reactome database was chosen. For advanced options, "only annotated genes" as statistical domain scope, "g:SCS threshold" as significance threshold and "0.05" as user threshold were selected.

## Visualization and interpretation of pathway enrichment analysis results

String output was uploaded to Cytoscape (https://cytoscape.org) (*Shannon et al., 2003*) version 3.9.1 in order to edit. Then, the result files of g:Profiler were uploaded to Cytoscape and a pathway network was constructed to visualize and interpret the output. Within the software, plug-in EnrichmentMap was used. According to Reimand et al. in the input panel of EnrichmentMap number of nodes section, FDR Q value cutoff was set to 0.01. For the number of edges, predefined parameters were used.

## Investigation of mutations

Mutation information of commonly mutated five genes (*TP53, PTEN, PIK3CA, IL7R*, and *KMT2D*) were retrieved from 16 articles that were included in this meta-analysis. Databases dbSNP (https://www.ncbi.nlm.nih.gov/snp/; *Sherry et al., 2001*), Catalogue of Somatic Mutations in Cancer (COSMIC) (https://cancer.sanger.ac.uk/cosmic; *Tate et al., 2019*), National Cancer Institute (NCI) Genomic Data Commons (GDC) data portal (https://portal.gdc.cancer.gov/; *Grossman et al., 2016*), and ClinVar (https://www.ncbi.nlm.nih.gov/clinvar; *Landrum et al., 2018*) were used to search the mutations.

## Drug search

National Institutes of Health (NIH) (https://www.nih.gov/; *National Cancer Institute, 2022*) was used to find FDA (Food and Drug Administration) approved drugs that treat NSCLC, breast cancer, melanoma, and brain tumor. Drug-Gene Interaction Database (DGIdb) (https://www.dgidb.org/) (*Freshour et al., 2021*) version v4.2.0—sha1 was used to find potential drugs that interact with selected genes. Clinicaltrials.gov (https://clinicaltrials.gov/; *U. S. National Library of Medicine, 2022*) database was used to search for present clinical status of candidate drugs.

## RESULTS

After thorough research, 221 studies were selected. There were no duplicate results as PubMed was used for the search. Among the 221 articles that were examined for their titles and abstracts, only 23 were qualified for a full-text review. Finally, 16 studies were included and assessed in this meta-analysis as shown in Fig. 1.
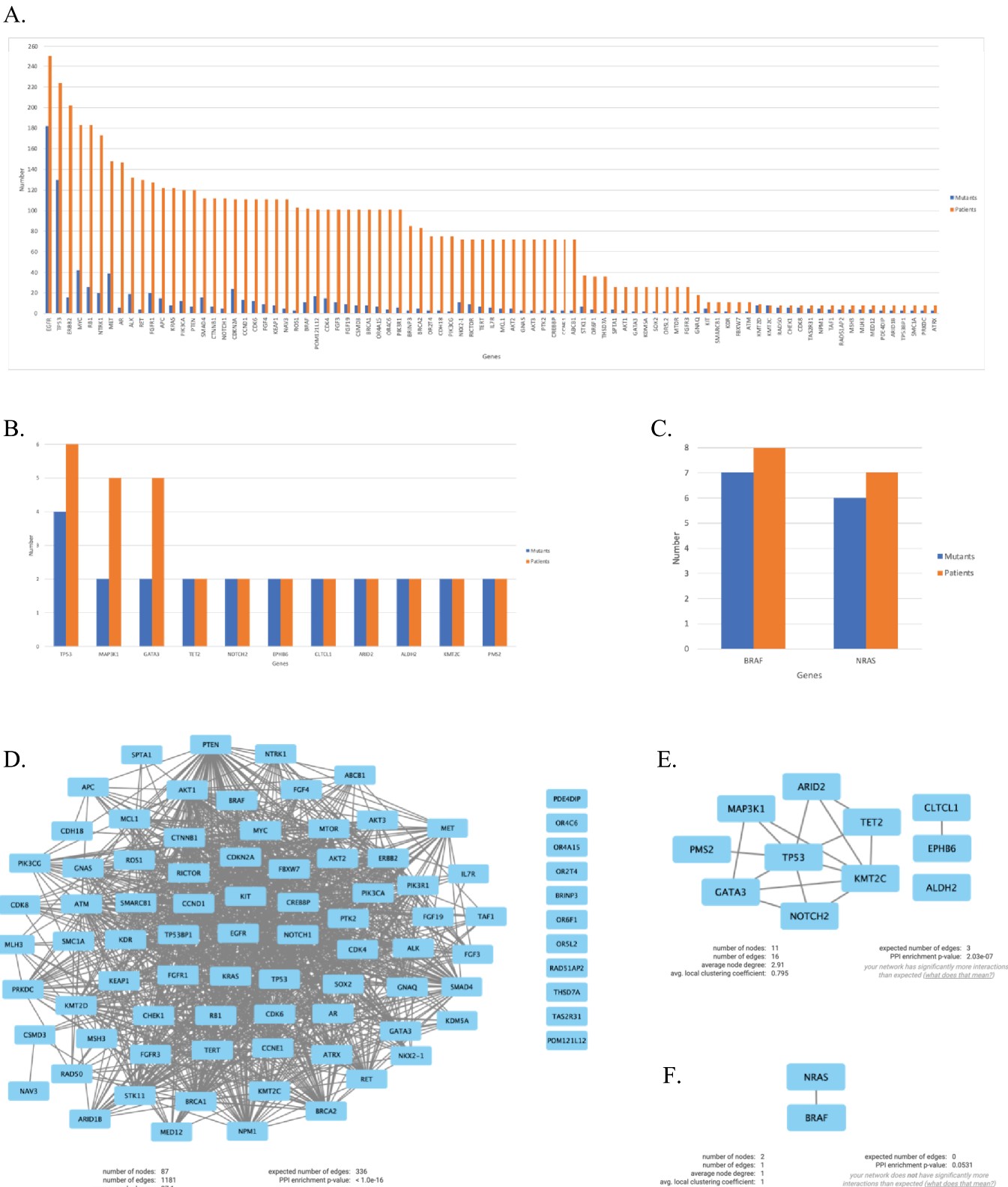

**Figure 4 Mutated genes and their interactions with each other in each primary cancer type.** Bar graph showing mutated genes in (A) NSCLC-LMC, (B) Breast-LMC and (C) Melanoma-LMC patients ($n \geq 2$). (D) PPI network of mutated genes in NSCLC-LMC. Among 87 proteins 12 of them

**Figure 4** (continued)
did not interact with any other proteins. (E) PPI network of mutated genes in breast-LMC. Among 11 proteins only one of them (ALDH2) did not make any interaction, while CLTCL1 and EPHB6 only interacted with each other. (F) PPI network of mutated genes in melanoma-LMC. NRAS and BRAF interacted with each other. PPI, proteinprotein interaction; LMC, leptomeningeal carcinomatosis; NSCLC, non-small cell lung cancer; *n*, patient number.

### Identification of mutated genes in LMC patients according to their primary tumor sites

In order to observe mutated genes according to primary tumor sites we created a venn diagram. Specific patient characteristic information such as age, gender, primary tumor mutation, treatment prior to LMC diagnosis, and treatment before sequencing were neglected because of the limited number of studies. Moreover, we wanted to focus on revealing the commonly mutated genes causing LMC out of the three primary cancers, NSCLC, breast cancer and melanoma. Hence, the current study includes mutated genes identified by NGS applied to CSF samples of LMC patients. A total of 96 genes were found to be mutated in all cancer groups, which distributed as 87 genes in NSCLC, 24 in breast and 13 in melanoma. 16 of these genes were shared between NSCLC and breast; 11 of them were common between NSCLC and melanoma; and six of them were shared between breast and melanoma. We found out that five genes (*TP53*, *PIK3CA*, *PTEN*, *IL7R* and *KMT2D*) were commonly mutated in each of the cancers (Fig. 3).

### PPI network construction of mutated genes in each primary cancer type

We continued our analysis with genes that are mutated in more than two patients for each primary tumor site, which are illustrated in Figs. 4A–4C. In order to investigate the potential interactions of genes mutated in NSCLC-LMC patients, we constructed a PPI network using STRING database. This PPI network contained 87 nodes and 1,181 edges (Fig. 4D). Eleven nodes and 16 edges were found in the PPI network of breast-LMC (Fig. 4E), and only two nodes and one edge were found in melanoma-LMC PPI network (Fig. 4F).

### PPI network and pathway enrichment analysis of commonly mutated genes in all three cancer groups

Next, we maintained our analysis with mutated genes that were observed in at least two primary tumor sites (Fig. 5A). We wondered if we could reveal a mechanism that causes metastasis from all three types of cancer to the meninges of the brain by investigating mutated genes. Therefore, we carried on our analysis with five genes that are commonly mutated in all three primary tumor sites. To see protein-protein interactions of TP53, PIK3CA, PTEN, IL7R and KMT2D, STRING database was used to construct a PPI network. PPI network included five nodes and six edges (Fig. 5B). Surprisingly, IL7R did not make an interaction with any other proteins. Next, we performed pathway enrichment analysis in order to investigate the pathways that these genes were involved in (Fig. 5C). The enriched GO pathways were presented in Fig. S1. There were seven pathways with some having subgroups.

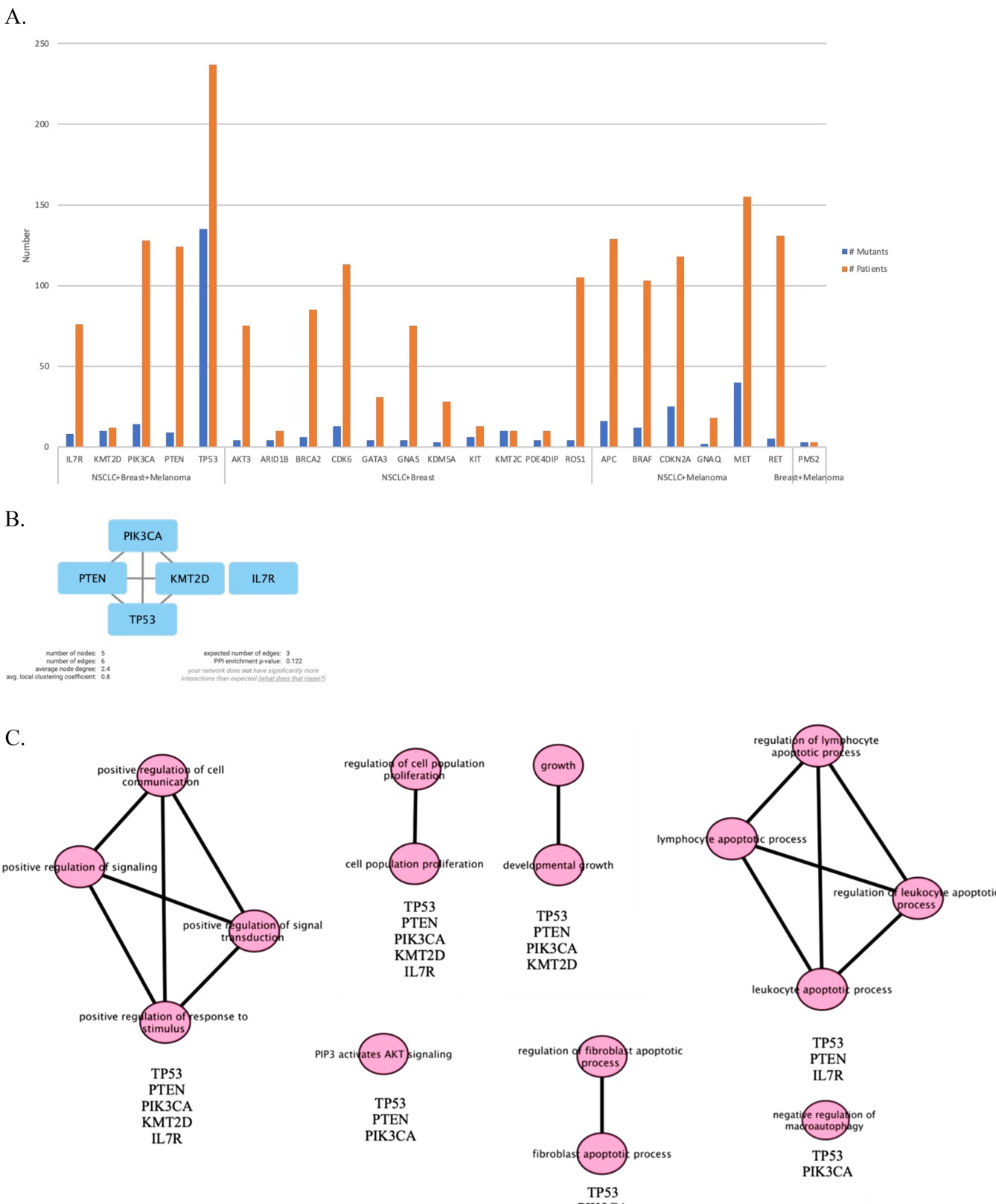

**Figure 5 PPI network and pathway enrichment analysis of commonly mutated genes in all three cancer groups.** (A) Bar-chart showing mutated genes in at least two primary tumor sites ($n \geq 2$). Five genes (*TP53, PTEN, PIK3CA, IL7R,* and *KMT2D*) were mutated in all three tumor

**Figure 5 (continued)**
sites. (B) PPI network of mutated genes appeared in all three cancer groups. All four proteins were interacted with each other, with the exception of IL7R. (C) Visualization of pathway enrichment analysis of mutated genes among all three cancer groups using Cytoscape. All five genes were involved in signaling and cell population proliferation pathways. TP53, PTEN, PIK3CA, and KMT2D had roles in pathways regarding growth. TP53, PTEN, and PIK3CA were involved in the AKT signaling pathway. TP53, PTEN, and IL7R were involved in lymphocyte and leukocyte apoptotic processes. Finally, TP53 and PIK3CA had roles in fibroblast apoptotic processes and regulation of macroautophagy. *n*, patient number; NSCLC, non-small cell lung cancer.               

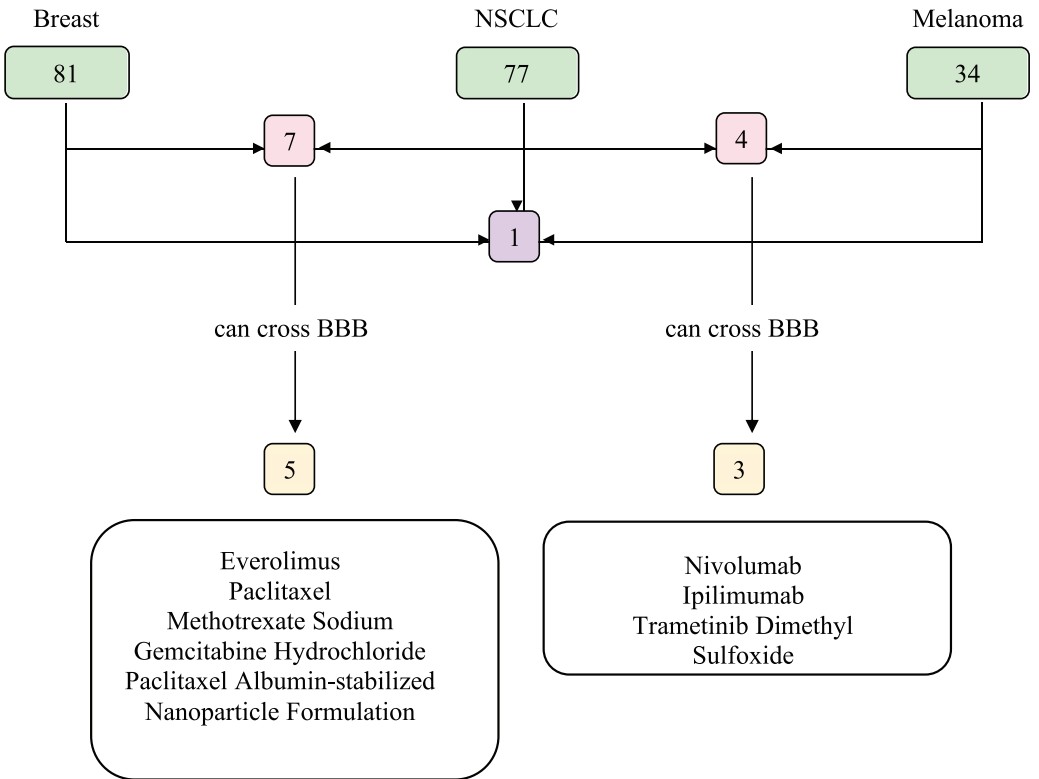

**Figure 6 Food and Drug Administration (FDA) approved drugs.** The number of FDA approved drugs used for treatments of breast cancer, NSCLC, and melanoma are shown in green. The number of drugs that are common in two cancer types are shown in pink and common in three cancer types shown in purple. Drugs able to cross BBB are shown in yellow. Five drugs that can cross BBB are shared between breast cancer and NSCLC patients, while three are common between NSCLC and melanoma patients. There are no common drugs between three cancer groups that can penetrate BBB. BBB, blood-brain barrier; NSCLC, non-small cell lung cancer.               

All five gene products were primarily associated with positive regulation of cell communication, positive regulation of signaling, positive regulation of response to stimulus, and positive regulation of signal transduction pathways. In addition, they are present in cell population proliferation and regulation of cell population proliferation pathways. In developmental growth and growth pathways TP53, PTEN, PIK3CA, and KMT2D were involved. TP53, PTEN, and IL7R had roles together in regulation of lymphocyte apoptotic process, lymphocyte apoptotic process, regulation of leukocyte apoptotic process, and leukocyte apoptotic process pathways. Whereas "PIP3 activates AKT signaling" pathway involved TP53, PTEN, and PIK3CA. Moreover, TP53 and

**Table 2 Drug candidates that can cross BBB for LMC treatment.**

| Drug candidate | Target gene | Clinical use | Clinical trials |
|---|---|---|---|
| Everolimus | *PTEN, PIK3CA* | NSCLC<br>Breast cancer | Melanoma |
| Bevacizumab | *PIK3CA, TP53* | NSCLC | NSCLC-LMC<br>Breast-LMC<br>Melanoma |
| Temozolomide | *PTEN, TP53* | – | Breast-LMC<br>NSCLC<br>Melanoma |

PIK3CA were present in regulation of fibroblast apoptotic process and fibroblast apoptotic process as well as negative regulation of macroautophagy.

## Investigation of mutations in commonly mutated genes

We extracted the mutation information of five commonly mutated genes in all primary tumor sites from articles included in this meta-analysis. Among 16 studies that were selected only 11 of them presented the mutation information of the genes. However, in these 11 studies, not all mutation information was reported. The mutation information of commonly mutated five genes can be accessed from File S4. We found 15 mutations in TP53, three mutations in PIK3CA and PTEN, two mutations in IL7R, and six mutations in KMT2D. We searched each mutation in dbSNP, COSMIC database, ClinVar, and GDC data portal. Some of these mutations were registered in the above-mentioned databases, whereas some were reported by only the articles that were analyzed in ours.

TP53 mutations R248Q and R248W are known as hotspot mutations, located in the DNA-binding domain of the TP53 gene. The R248Q mutation results in a diminished binding of DNA and a decrease in the activation of its targets, which hinders the transactivation of wild-type TP53 and leads to resistance against apoptosis and G1 arrest in cell culture, as evidenced by several research articles (*Hanel et al., 2013*; *Dearth et al., 2007*; *Boettcher et al., 2019*). R248Q was seen in four NSCLC-LMC and melanoma-LMC patients, while according to the GDC data portal there were eight breast and six lung cancer NCI cases that were registered. On the other hand, the R248W mutation was shown to reduce the transactivation of TP53 targets, promote cell proliferation in culture, and lead to an increase in tumorigenesis in mice, which was accompanied by decreased activation of ATM and an increase in genetic instability (*Song, Hollstein & Xu, 2007*; *Willis et al., 2004*). Although the R248W mutation was observed in only one NSCLC-LMC patient, it was found in nine breast, five lung, and three skin cancer NCI cases registered in GDC data portal. Another hotspot mutation among LMC patients was PIK3CA E545K, which is located in the PIK helical domain and results in increased AKT and MEK1/2 phosphorylation, cell survival independent of growth factors, and cell transformation in culture (*Dogruluk et al., 2015*; *Ng et al., 2018*). Six NSCLC-LMC and melanoma-LMC patients had this mutation, whereas 83 breast, 23 lung, and two skin cancer NCI cases were registered. Two KMT2D mutations, Q3278del and Q3861del, were seen in six

NSCLC-LMC patients, however none of these mutations were registered in any mutation database.

## Drug search

We investigated the drugs that could be used in common to target mutated genes in all three types of cancer. Therefore, FDA approved drugs that treat NSCLC, breast cancer, melanoma, and brain tumors were found with NIH (Fig. 6). Among these, drugs that have capability to penetrate the blood-brain barrier (BBB) to reach tumor site were selected and searched in DGIdb to look for interaction with the selected genes (*TP53, PIK3CA, PTEN, IL7R* and *KMT2D*). We then used Clinicaltrials.gov (https://clinicaltrials.gov/) database to search for present clinical status of candidate drugs. While no drugs were found to interact with IL7R and KMT2D, three drug candidates, Everolimus, Bevacizumab and Temozolomide, had interactions with TP53, PIK3CA, and PTEN, which are listed in Table 2.

## DISCUSSION

LMC is a life-threatening consequence of solid tumors that has a poor prognosis regardless of treatment (*Pellerino et al., 2020*). Because effective treatment strategies against primary tumors are developing every day, LMC incidences are increasing (*Lee et al., 2013*). Although the number of sequencing applications to LMC patients has been increasing in recent years, they are mainly aimed at facilitating the diagnosis of the disease. There are limited studies that investigated differentially expressed genes (DEGs) in LMC patients *in vitro*. One study revealed *CEACAM6* gene, which had elevated expression and correlation with cellular migration in NSCLC cell line (*Li et al., 2021*). Therefore, it is vital to establish biomarkers and the exact molecular mechanism for the prognosis and treatment of LMC. In this study, mutated genes of three types of primary cancer that caused LMC were analyzed with bioinformatics tools to determine related pathways which may be involved in the development of LMC.

According to this meta-analysis, *TP53, PIK3CA*, *PTEN*, *KMT2D*, and *IL7R* genes were commonly mutated in LMC patients caused by NSCLC, breast, and melanoma. The PPI network constructed with STRING showed that there is an interaction within these gene products except IL7R. Even though IL7R lacked direct interaction and stayed alone, it was observed that IL7R was involved in the enriched pathways with other four genes. According to the pathway enrichment analysis performed with g:Profiler, apoptotic processes of lymphocytes and leukocytes, regulation of signaling pathways, growth, fibroblast apoptotic process, proliferation of cell population, AKT signaling and regulation of macroautophagy were found to be relevant with occurrence of LMC. These findings also pointed to important clues about molecular interactions in LMC development.

All five genes were involved in both regulation of signaling and cell population proliferation pathways. Signaling pathways are responsible for many vital characteristics of cells like cell-to-cell communication, cell proliferation, cell growth, cell death, and cellular movement. In normal cells, cell proliferation and cell-to cell communication are tightly regulated by PI3K-AKT and RAS-ERK pathways. Due to any alterations in one of these

five genes, signaling pathways may act on cancer progression, in which cancer cells will excessively proliferate (*Sever & Brugge, 2015*). One of these genes, PIK3CA encodes the p110α subunit of Class IA PI3K, and the other PTEN negatively regulates the PI3K pathway by acting as an antagonist (*Yang et al., 2019*). Aberrant PI3K signaling, loss of function mutations of *PTEN*, and *PI3K* mutations have the ability to promote invasion, metastasis and immune cell modulation in brain metastasis (*Crespo, Kind & Arcaro, 2016*). *PIK3CA* mutations are among the most frequently seen genetic changes in breast cancer, occurring in about 25–45% of cases. Studies have revealed that *PIK3CA* mutations are more frequent in estrogen receptor positive breast cancer (*Samuels & Waldman, 2010*). E545K is one of the most common alterations of *PIK3CA* seen in breast cancer patients (*Fusco et al., 2021*). *PTEN* is another most frequently changed gene in breast cancer. In 5–10% of breast cancer patients, frameshift mutations cause loss of function of PTEN (*Carbognin et al., 2019*). In NSCLC, the PI3K/Akt/mTOR pathway has been heavily implicated in both tumorigenesis and the progression of disease. Somatic mutations of *PIK3CA* and amplifications of *PIK3CA* are frequently found in patients with NSCLC. According to a large cohort study, certain NSCLC subtypes, such squamous cell carcinoma, exhibit a higher prevalence of *PIK3CA* mutations than others. Among those, the most prevalent mutation was E545K exon 9 (57.1%) (*Tan, 2020*). *PTEN* genetic changes are seen in lung cancer at a low frequency (2–7%). The prevalence of PTEN protein loss is significantly higher compared to the modest frequency of genetic changes. It has been reported in more than 40% of NSCLC cases (*Gkountakos et al., 2019*). In another cohort study, sequencing findings suggested that PI3K pathway abnormalities may be linked to an increased risk of developing NSCLC-LMC (*Fan et al., 2018*). Research has demonstrated that patients with melanoma may develop acquired resistance to anti-cancer therapy due to the activation of the PI3K signaling pathway, either as a result of *PTEN* loss or *PIK3CA* mutations (*Tran et al., 2021*). *PTEN* has been demonstrated to be inactivated or deleted in up to one-third of melanomas (*Dong et al., 2014*).

In addition to these, tumor suppressor TP53, halts cell proliferation in response to various stress conditions such as DNA damage and is also involved in DNA repair, cell cycle, cell metabolism, apoptosis, senescence, and autophagy (*Liang, 2010*). *TP53* gene mutations are prevalent in breast cancer, occurring in 20–30% of cases, and have been associated with more aggressive subtypes of the disease, as evidenced by studies (*Shahbandi, Nguyen & Jackson, 2020*). Truncating mutations in TP53 can potentially promote tumor growth. For example, the TP53 exon 6 truncating mutant R196* has been shown to facilitate tumor cell growth and metastasis (*Chen et al., 2022*). *TP53* gene mutations are found in approximately 50% of cases of NSCLC (*Mogi & Kuwano, 2011*). Studies have suggested that *TP53* mutations are associated with a poor prognosis in NSCLC, as they can disrupt the normal function of the TP53 protein (*Chen et al., 2022*). It was reported that TP53 mutations are found in approximately 15–25% of melanoma cases, which are linked to a higher risk of recurrence and shorter overall survival (*DeLeon et al., 2020*). These mutations may also play a crucial role in the development of metastatic melanoma (*Loureiro et al., 2020*). In the concept of cell survival and death, interaction of PI3K/AKT pathway and TP53 have substantial roles. Several studies indicated that TP53

has the ability to activate PTEN by influencing PI3K (*Abraham & O'Neill, 2014*). A variety of different mechanisms also stimulate cell proliferation. PI3K/AKT and JAK/STAT pathways can be triggered by IL7R, which is the receptor of IL7, an important cytokine. In addition to that, IL7 plays a fundamental role in development of lymphocytes (*Campos, Pissinato & Yunes, 2019*). Tumor infiltrating cells (TILs) that are found in the tumor microenvironment inhibit progression of cancer and are successful in active cancer immunotherapy (*Zhang et al., 2019*). IL7 is a proper target for improving immune system function. It has the ability to re-establish the immune system, enhance T cell function *in vivo*, and oppose the immune system's suppressive network (*Gao et al., 2015*). Epigenetic mechanisms also regulate cell proliferation and survival. KMT2D, which is a histone methyltransferase catalyzes the specific lysine residues on histones and proteins. It is deleted and mutated in a variety of cancers, primarily melanoma and lung cancer, then breast cancer (*Dhar & Lee, 2021*). KMT2D takes part in the TP53 and PI3K pathways as an epigenetic modifier that is responsible for stimulating gene expression (*Lv et al., 2018*). Mutations in *KMT2D* suggest epigenetic mechanisms may be involved in LMC development whereas it is not very surprising to see *TP53*, *PIK3CA*, and *PTEN* gene mutations since they have roles in multiple pathways in our body and frequently mutated in multiple cancer types.

According to pathway enrichment analysis, TP53, PTEN, PIK3CA and KMT2D also take place in pathways related with growth. Uncontrolled cell growth may occur due to any mutations according to tumor type. With the presence of cytokines and growth factors many apoptotic pathways can have an effect on the tumor microenvironment to form an anti-tumor response (*Zhu, Petit & Van den Eynde, 2019*; *Anderson & Simon, 2020*). The other two pathway classes consist of regulation of leukocyte and lymphocyte apoptotic processes and fibroblast apoptotic processes. Apoptosis, which is controlled cell death, is important for immune function. Strict regulation of apoptosis is vital for lymphocytes and leukocytes, as the altered regulation and mutations in *TP53*, *PTEN*, *IL7R* genes can result in cancer formation (*Zhu, Petit & Van den Eynde, 2019*; *Carrero & Unanue, 2006*). Progression of cancer is triggered by the intense relation of stromal cells found in the tumor microenvironment. Moreover, the last enriched pathway involved the negative regulation of macroautophagy. Macroautophagy, also referred as autophagy, is important for the breakdown of old proteins and damaged organelles, as well as keeping balance of the cellular environment. Autophagy can facilitate tumorigenesis by contributing to cancer-cell proliferation, but it also acts as a tumor suppressor by inducing apoptosis. Our analysis showed that TP53 and PIK3CA negatively regulate macroautophagy. Although PTEN was not involved in this pathway according to our analysis, several studies showed that PTEN and TP53 alterations are frequently observed in brain tumors and have roles in regulation of autophagy (*Kaza, Kohli & Roth, 2012*). However, it remains unclear whether it contributes to tumor progression or suppression due to its dual effects in LMC (*Shi, Norberg & Vakifahmetoglu-Norberg, 2021*; *Yun & Lee, 2018*).

Further investigation of mutations in TP53, PIK3CA, PTEN, KMT2D, and IL7R were also performed in this study. Three hotspot mutations were seen in LMC patients, TP53 R248Q, TP53 248W, and PIK3CA E545K. In a study consisting of 23 cases of CNS

metastatic breast cancer, high frequency of TP53 mutations were detected. Those mutations included R248Q, R248W, and R196* (*Lo Nigro et al., 2012*). In one research article, PIK3CA gene mutations, such as E545K were found in NSCLC-CNS metastatic lesions (*Nicoś et al., 2016*). Recent studies demonstrated that ER+/HER2-breast cancer patients may be more likely to develop CNS metastasis if they have PIK3CA-activating mutations. Brain metastases were substantially more frequent in patients with PIK3CA mutations, such as E545K among 307 patients with ER+/HER2 metastatic breast cancer (*Batalini et al., 2020*). Although some of the mutations discussed in this meta-analysis have some proof to induce brain metastasis, these mutations that were investigated in this research study was very limited to their availability in the articles that formed the basis of this study. There could still be many different mutations in the five genes revealed in this study and their togetherness could yet be important for the development of LMC. That is why, the effect the mutational differences/commonalities between these genes represented in this article could further be investigated in future studies. Moreover, it will be groundbreaking if any signature mutations were discovered to identify LMC which would open the gates for developing new treatment opportunities.

In recent years, molecular targeted cancer therapy is in demand. NSCLC, breast cancer and melanoma, which are primary tumor types of LMC may have marker mutations such as *EGFR*, *HER2* and *BRAF* respectively. There are targeted therapy options for these kinds of mutations. Unfortunately, there is still no gold standard in treatment of LMC. Current management of this disease includes radiotherapy, systemic and intrathecal chemotherapy that is commonly described in the literature as cytarabine and methotrexate (*Lee et al., 2016*). One concept that should be kept in mind when treating LMC is that the drugs must penetrate BBB to reach the tumor site.

Our analysis revealed that, among 179 FDA approved chemotherapy drug options for NSCLC, breast cancer, and melanoma, only one of them was common but it could not cross BBB. There were a limited number of BBB-penetrating drug options for NSCLC-breast cancer and NSCLC-melanoma. To find out drugs that could both penetrate BBB and target these five genes, FDA approved brain tumor drugs were examined. Among seven drugs, Everolimus, Bevacizumab, and Temozolomide that target PTEN, PIK3CA, and TP53 respectively are revealed as targeted therapy candidates for LMC treatment with patients bearing mutations in any of these genes. In addition to these, there are many drugs that are under clinical investigation for NSCLC, breast cancer and melanoma that target altered genes. Disulfiram that targets TP53 and Buparlisib that target PIK3CA and PTEN can be given as examples. Hopefully, in the near future clinical trials involving differential use of such drugs can shed light to treatment of LMC.

During the study some limitations were encountered. As this meta-analysis was composed of publicly available articles, it also limited the availability of the sequencing information about LMC. Because LMC is not that frequently observed, cancer databases like The Cancer Genome Atlas (TCGA) have no information about LMC that can be further analyzed. The other is that the information about mutations of patients' primary tumor sites was inadequate. Moreover, effects of commonly mutated genes were not further explored due to lack of experimental research. Despite these limitations, we hope

that this study will be a pioneer in future studies of LMC and that this lack of information will be eliminated.

## CONCLUSIONS

In conclusion, a total of 96 mutated genes in LMC were investigated *via* integrated bioinformatic analysis to further understand the molecular mechanism of LMC. Considering the vital roles of genes commonly mutated in each primary tumor type, further research into their precise pathways in the tumorigenesis and prognosis of LMC, particularly TP53, PTEN, PIK3CA, KMT2D, and IL7R, may be pursued. Due to unknown aspects of the molecular mechanism of LMC, larger cohort studies with multiple omics approaches, and the development of new drug candidates and combinatorial drug trials, are much needed.

### Funding
The authors received no funding for this work.

### Competing Interests
The authors declare that they have no competing interests.

### Author Contributions
- Irem Congur conceived and designed the experiments, performed the experiments, analyzed the data, prepared figures and/or tables, authored or reviewed drafts of the article, and approved the final draft.
- Ekin Koni conceived and designed the experiments, performed the experiments, analyzed the data, prepared figures and/or tables, authored or reviewed drafts of the article, and approved the final draft.
- Onur Emre Onat analyzed the data, prepared figures and/or tables, authored or reviewed drafts of the article, and approved the final draft.
- Zeynep Tokcaer Keskin conceived and designed the experiments, analyzed the data, authored or reviewed drafts of the article, and approved the final draft.

### Data Availability
This raw data is available in the Supplemental File.

### Supplemental Information
Supplemental information for this article can be found online at http://dx.doi.org/10.7717/peerj.15250#supplemental-information.

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
