# Peer review of "Meta-analysis of commonly mutated genes in leptomeningeal carcinomatosis"

_PeerJ, doi:10.7717/peerj.15250_

## Round 0.1 · original submission · Major Revisions

Dear Dr. Tokcaer Keskin,

Thank you for your submission to PeerJ.

It is my opinion as the Academic Editor for your article - Meta-analysis of commonly mutated genes in leptomeningeal carcinomatosis - that it requires a number of Major Revisions.

With kind regards,
Abhishek Tyagi
Academic Editor
PeerJ Life & Environment

Reviewer 1 ·

Basic reporting

The authors aimed to identify commonly mutated genes in leptomeningeal carcinomatosis (LMC) caused by non-small cell lung cancer (NSCLC), breast cancer, and melanoma using an in-silico approach and their interactions using integrated bioinformatic approaches and tools in meta-analysis.

The authors found TP53, PTEN, PIK3CA, KMT2D, and IL7R are likely to have vital roles in LMC development.

The language used in this manuscript is clear, unambiguous and professional. Literature references and sufficient background are provided. However, if the author can provide more molecular biology background of TP53, PTEN, PIK3CA, KMT2D, and IL7R in the figures, it would be easier for the reader to understand.

It would be better if the authors can synchronize the font in different figures.

Some small mistakes are found in the manuscript.
Line 41 "in-slico" the authors may mean "in-silico"

Experimental design

The experiments were designed properly. The research questions were well-defined, relevant, and meaningful. The methods were also described in sufficient detail.

Validity of the findings

The authors pointed out the potential importance of TP53, PTEN, PIK3CA, KMT2D, and IL7R in LMC development and suggested more research should be done to explain the mechanism beyond.

Reviewer 2 ·

Basic reporting

The manuscript entitled ‘Meta-analysis of commonly mutated genes in
leptomeningeal carcinomatosis’ reports a meta-analysis identified the potential mutated genes including TP53, PTEN, PIK3CA, KMT2D, and IL7R, in leptomeningeal carcinomatosis originated from different tumor sites. The methods for study selection and data extraction are rigorous and clearly and thoroughly presented. Though the conclusions seem to be valuable, I am not fully convinced by some details in the manuscript (please refer to the comment under Validity of the findings).

Experimental design

no comment

Validity of the findings

Major comment: Instead of the mutated genes found and listed in this paper, the specific variant/mutation would be more interesting to report. For example, PIK3CA p.E545K is shared in Li Y et al. 2016 (breast cancer) and Ballester et al. 2018 (melanoma). Conventional EGFR mutations (T790M, 19 del, 20 ins, L858R) were reported across NSCLC patients. It would be nice if the authors commented on specific mutations, which would bring the key findings from the meta-analysis to the next level.

Additional comments

no comment

---

## Round 0.2 · accepted · Accept

Dear Dr. Tokcaer Keskin,

Thank you for your submission to PeerJ.

I am writing to inform you that your manuscript - Meta-analysis of commonly mutated genes in leptomeningeal carcinomatosis - has been Accepted for publication.

Congratulations!

Reviewer 1 ·

Basic reporting

The authors edited the manuscript properly. However, details like fonts can still be improved.

Experimental design

No comment.

Validity of the findings

No comment.

Reviewer 2 ·

Basic reporting

no comment

Experimental design

no comment

Validity of the findings

no comment

Additional comments

The authors have addressed my concrens and thus I recommend the acceptance of this manuscript.